# Regression Models for In Vivo Discrimination of the Iberian Pig Feeding Regime after Near Infrared Spectroscopy Analysis of Faeces

**DOI:** 10.3390/ani14111548

**Published:** 2024-05-24

**Authors:** Pablo Rodríguez-Hernández, Vicente Rodríguez-Estévez, Cristina Burguillo-Martín, Nieves Núñez-Sánchez

**Affiliations:** Departamento de Producción Animal, Facultad de Veterinaria, Universidad de Córdoba, Campus de Rabanales, 14071 Córdoba, Spain; v22rohep@uco.es (P.R.-H.); crisburmar1208@gmail.com (C.B.-M.); pa2nusan@uco.es (N.N.-S.)

**Keywords:** Iberian pig, authentication, spectral data, grazing diet, feeding regime prediction, chemometric models, animal production

## Abstract

**Simple Summary:**

Nowadays, there is a growing demand among consumers for products sourced from animals raised on a natural diet, relying solely on grazed natural resources. This is the case for Iberian pigs that forage for acorns and grass without any additional feed. However, current regulation does not consider any official analytical methodology for the discrimination and certification of their feeding regime. Although this objective was pursued in the past, it was performed by evaluating postmortem samples. This new research evaluates the potential of Near Infrared Spectroscopy (NIRS) to discriminate the diet of Iberian pigs in vivo, using spectral information from faecal samples and regression analysis. The final models demonstrated a robust performance in cross-validation, also achieving 94% prediction accuracy. These results suggest that NIRS analysis of faecal samples can be a valuable approach for achieving the objective above-mentioned and providing useful information for in vivo discrimination of diets in Iberian pig production.

**Abstract:**

The Iberian pig is a native breed of the Iberian Peninsula, which holds an international reputation due to the superior quality and the added value of its products. Different rearing practices and feeding regimes are regulated, resulting in different labelling schemes. However, there is no official analytical methodology that is standardised for certification purposes in the sector. Near Infrared Spectroscopy (NIRS) is a technology that provides information about the physicochemical composition of a sample, with several advantages that have enabled its implementation in different fields. Although it has already been successfully used for the analysis of Iberian pig’s final products, samples evaluated with NIRS technology are characterised by a postmortem collection. The goal of this study was to evaluate the potential of NIRS analysis of faeces for in vivo discrimination of the Iberian pig feeding regime, using the spectral information per se for the development of modified partial least squares regressions. Faecal samples were used due to their easy collection, especially in extensive systems where pig handling is difficult. A total of 166 individual samples were collected from 12 farms, where the three different feeding regimes available in the sector were ensured. Although slight differences were detected depending on the chemometric approach, the best models obtained a classification success and a prediction accuracy of over 94% for feeding regime discrimination. The results are considered very satisfactory and suggest NIRS analysis of faeces as a promising approach for the in vivo discrimination of the Iberian pigs’ diet, and its implementation during field inspections, a significative achievement for the sector.

## 1. Introduction

The Iberian pig constitutes a native breed traditionally reared in the southwest of the Iberian Peninsula, owing to its good ability to adapt to the environment and the feeding resources available in the *dehesa* agroecosystem [1,2]. The different products obtained from this breed hold an important international reputation and are appreciated worldwide due to their quality and their added value production systems [3,4]. Despite the relation between the *dehesa* agroecosystem and this native breed [2], the current production systems in the Iberian pig sector have important differences that mainly depend on animal diet and management, from extensive to intensive systems. 

The current legal framework of the Iberian pig sector is available at Real Decreto 4/2014, which summarises quality and productive conditions that must be ensured throughout the whole process of Iberian pig production, including labelling, commercialisation, traceability, and certification, from farm to finished products [5,6]. In this regulation, the following three different commercial categories are defined in terms of animal feeding and management: acorn-fed or “bellota” (AF), totally extensive systems placed in the *dehesa* agroecosystem; outdoors feed-fed or “cebo de campo” (OFF), which can be considered as mixed systems; and lastly, intensive feed-fed or “cebo” (IFF), which refers to intensive systems where animals are completely stabled without access to free range [5]. Thus, important differences in the products’ final price are obtained in these categories, as diet and management are considered the main factors in determining its quality [7]. 

At present, the certification procedures established by Real Decreto 4/2014 regarding the feeding regime and the production systems in the Iberian pig sector must be carried out by accredited companies [5]. In practice, these tasks are based on visual and documentary inspections carried out by technicians visiting Iberian pig farms. This fact has often been criticised as visual control of animals’ diet can be subjective and insufficient, especially considering the price and reputation of products coming from Iberian pig production. Although there have been attempts in the past, there is currently no standardised analytical methodology for certifying the pig feeding regime. In addition, considering increasing consumer awareness of food origin and labelling [8], the inclusion of analytical techniques within the certification procedures of the sector is considered essential to assure reliable information to consumers and to protect producers from fraudulent practices [9,10]. Against that background, a great deal of research focused on the development and evaluation of this type of techniques has been published in recent years in the sector, such as methodologies based on liquid and gas chromatography or different spectroscopic techniques.

Near Infrared Spectroscopy (NIRS) is a technology based on the correlation between the physicochemical characteristics of a sample and the radiation absorption in the infrared region [11]. Several advantages such as the ease of use, a competitive price, the possibility of providing qualitative and quantitative information with a single and non-destructive analysis, as well as its reproducibility and repeatability have induced the implementation of NIRS in the agri-food industry [12]. This technology has demonstrated great versatility by analysing different type of samples, including Iberian pig products, although these have usually been food samples or carcasses [13,14,15], all characterised by a postmortem collection. Other samples such as faeces have received limited attention. However, the great deal of information that it provides about the internal metabolism as well as its simple collection are regarded as notable strengths [16]. Therefore, the analysis of this type of sample might provide an innovative and interesting approach in Iberian pig production, as it would allow an in vivo analysis. Pig faeces have been successfully analysed with NIRS technology before, although the available studies are mainly focused on digestibility evaluation and other objectives [17,18,19]. 

Thus, the aim of the present study was to evaluate the interest of NIRS analysis of faecal samples to discriminate the Iberian pig feeding regime; chemometric multivariate analysis of the resulting spectral data were applied to perform Modified Partial Least Squares (MPLS) regressions. 

## 2. Materials

### 2.1. Animals, Feeding Regimes and Sample Collection

A total of 166 Iberian pigs from 12 different farms over three consecutive years were included in the study. Farms were located in the provinces of Cordoba and Huelva (Southern Spain, Andalucia). The animals evaluated were fattening Iberian pigs in the last stage of the productive cycle, from approximately 100 to 160 kg live weight. The pigs were raised within mixed groups of pure and crossbred male and female castrated Iberian pigs at 16–18 months of age. Breed purity of animals varied from 100% to 50% Iberian breed depending on the farm strategy due to crossbreeding with the Duroc breed, as allowed in regulation [5].

For the study design, the farms included were allocated into two groups based on their feeding regime and finishing production system, according to the current regulation [5]. A first group of extensive farms belonged to AF commercial category, which included grazing animals that were exclusively fed with natural resources, mainly acorns and grass in the *dehesa*, without using any supplementary feed [2]. This type of system is called *montanera* and constitutes the origin of acorn hams, the most famous products worldwide coming from the Iberian pig. On the other hand, a second group of farms were identified as feed-fed (FF), where animals fed with a diet based on concentrate feeds were included. The two remaining commercial categories available in the market were included in this group (FF): OFF farms, mixed production systems where animals were raised outdoors with a feeding regime mainly depending on feed and sporadic access to natural resources; and IFF farms, where pigs were permanently stabled and fed with compound feed [5]. The difference between both basically involved the rearing system as follows: while OFF animals had access to outdoor facilities and could forage grass and other resources, IFF pigs were reared under controlled conditions in closed buildings, similarly to conventional intensive white pig production. These categories are in line with the current commercial labels established by official regulation (Real Decreto 4/2014) and farms included in the study were officially certified [5]. Animals were exclusively fed with the same feeding regime for at least 1.5 months before sample collection. The distribution of animals and feeding regimes was as follows: 93 animals from 7 farms for the AF category and 73 animals from 5 farms for the FF category. 

Regarding the composition and nutritional value of diets compared in the present study, on the one hand, the compound feeds used on OFF and IFF farms were based on different cereals, animal and vegetal fats, oil seeds and minerals as the main components. The mean average composition of feed used on farms was as follows: 10.82–13.72% crude protein, 3.23–7.64% crude fibre, 3.6–7.54% crude fat, 4.10–7.99% crude ash, 0.56–0.73% lysine, 0.16–0.24% methionine, 0.44–1.67% calcium, 0.35–0.7% phosphorus, and 0.13–0.26% sodium. On the other hand, and as mentioned above, the diet on AF farms was based on acorn and grass, although other natural resources could also be grazed by pigs in the *dehesa*, such as berries, roots, bushes, or mushrooms [2]. The natural and diverse character of this diet substantially hampers the determination of a mean average composition, but an estimation about chemical composition of acorn and grass grazed by Iberian pigs during the *montanera* was published a few years ago [2,20]. 

A total of 166 fresh faecal samples were individually collected from all the animals in the 12 farms included in the study. Samples were collected after spontaneous defaecation, without any animal restraint or stress, directly from the ground. To do this, pigs were continuously monitored until defaecation, to assure individual traceability of samples. Collection of faecal samples took approximately one morning per farm, as defaecation was not simultaneous for the whole group of evaluated animals. During collection, the part of faeces in contact with the soil was discarded to avoid the inclusion of other materials or possible contaminations. Once collected, faecal samples were refrigerated in boxes and transported to the laboratory, where they were stored at −18 °C until processing and NIRS analysis. This collection process of the faecal samples and its subsequent management has already been used with success in previous studies with similar objectives in Iberian pig production [6,21].

### 2.2. Sample Processing and NIRS Analysis

Firstly, the processing of faecal samples consisted in an overnight thawing procedure on the day prior to NIRS analysis. Then, the samples were oven-dried for 48 h at 60 °C, and lastly, milled to pass through a 1 mm sieve. The equipment used for faeces analysis in the study was a FOSS-NIRSystem 6500 scanning monochromator (FOSS-NIRSystems, Silver Spring, MD, USA) equipped with a transport module. Spectral absorbance data of samples were recorded in reflectance mode from 400 to 2500 nanometres (nm), every 2 nm, as log 1/R, where R was the sample reflectance. Each spectrum was time-averaged from 32 scans and compared with the 16 measurements of a ceramic reference. Spectra of the faecal samples were acquired using a small ring cup and one spectrum per sample was obtained. The methodology described here for the faecal samples has been widely used in previous studies with other animal species such as sheep, goats or rabbits [22,23,24].

### 2.3. Chemometrics

MATLAB^®^ environment (The Mathworks Inc., Natick, MA, USA, 2007), its PLS Toolbox software (Eigenvector Research, Inc., Manson, WA, USA), and WinISI IV software package (version 4.8, Foss, Hillerød, Denmark) were employed for chemometric treatment of the spectral data. 

Firstly, the mean spectra were represented to evaluate potential differences between the two groups compared. Also, a principal component analysis (PCA) was performed as an exploratory and cluster analysis method to notice potential trends within spectral data and detect possible outliers. Then, the dummy regression technique [25] was used to perform MPLS regression models and therefore predict the pigs’ feeding regime. For that purpose, a categorical dummy Y variable was created, and samples were assigned a value of the dummy variable according to the group in which they belonged: 1 for AF category and 2 for FF category. The initial set of samples (n = 166) was split into a training set, which included 80% of the total (n = 133) for calibration and cross-validation, and a prediction or validation set with the remaining 20% of the samples (n = 33), which were used as blind or unknown samples for the external validation of regressions. The following three different spectral regions were compared: the whole spectrum (400–2500 nm), near infrared region (850–2500 nm), and the segment (400–850 nm) including the spectrum visible region. 

Mathematical pre-treatments using the first or second derivative with and without a scatter correction were performed. These spectral derivatives were used to remove additive and multiplicative effects in the resulting spectra of the faecal samples [11]. These were named using four numbers as follows: the first digit regarding the derivative order (1 or 2 depending on first or second derivative, respectively); the second one regarding the derivation segment (5), as an interval expressed in nm which is used for the calculation of the derivative; and the third and fourth digits referring to the smoothing segments (5 and 1, respectively) as intervals expressed in nm for smoothing calculation. Standard normal variate and detrending algorithms were employed to correct the scatter effect [26]. MPLS regressions were also developed without any pre-treatment; thus, five models were obtained for each spectral region evaluated (400–2500 nm, 850–2500 nm and 400–850 nm): 1, 5, 5, 1 and 2, 5, 5, 1, both with and without scatter correction, as well as a MPLS regression model using the raw spectral data.

Samples were classified as AF or FF when the predicted value was equal to or within ±0.5 of the corresponding dummy value. The standard error of calibration (SEC), cross-validation (SECV) and prediction (SEP), the classification success and the coefficient of determination in calibration, cross-validation, and prediction (R^2^_C_, R^2^_CV_ and R^2^_P_, respectively) were the statistics calculated for each model. The best regression in each spectral region was selected on the basis of the lower SEP and SECV, higher R^2^_P_ and R^2^_CV_ values, and classification success. 

## 3. Results and Discussion

NIRS analysis of the faeces resulted in a final database with over 170,000 pieces of data. It consisted of 1050 columns, corresponding to the different wavelengths provided by the NIRS equipment, and 166 rows, representing the individual faecal samples obtained from Iberian pigs. A slightly higher number of AF animals and farms were included in the study design due to the heterogeneity existing in this kind of extensive system in terms of the natural resources grazed by the pigs and therefore their final diets [2]. The final goal in this regard was to cover as much variability as possible. 

The mean spectra comparison of the two groups highlighted significant differences over the entire spectrum (Figure 1). In general, a higher mean absorbance was obtained for the AF group in comparison with the mean spectrum coming from pigs included in the FF group (Figure 1). The different absorbance data could be related to changes in faeces composition, and in the concentration and presence of certain biological components; these findings were probably induced by the feeding regimes compared. In this regard, differences between absorption bands corresponding to protein, fat and fibre content were noticed: bands at 1516, 2056, 2174, 2468 nm corresponded to the region where proteins are absorbed [27]; bands around 1210, 1726, 1760, 2308, 2348 nm corresponded to the absorption of fatty acids and fat content [28]; and bands at 1922, 2078–2110, 2268, 2420 nm corresponded to fibre content [29] (Figure 1). This variability in faeces composition has been previously highlighted using NIRS analysis in other animals’ species such as rabbits, and it has been related not only to the diet composition given to animals, but also to the interaction of this with the digestive physiology of individuals [24].

Afterwards, the PCA revealed no outliers or atypical samples within the entire set of samples (166), so the whole set of samples was used to develop MPLS regressions. Also, an interesting finding during the observation of its graphic representation was made: although PCA is not defined as a discriminant analysis, a certain separation between the two groups compared was noticed. This preliminary tendency can be observed in the PCA score plot (Figure 2), yet some overlapping between the two groups was observed. Interestingly, samples from the FF group (in green) which were overlapped with the AF samples (in red) belonged to the OFF subcategory. The above-mentioned overlapping between the AF and OFF groups may be explained considering these two feeding strategies: both groups of animals have access to the outdoors, and although OFF pigs are mainly fed with concentrate feed, they may have sporadic access to outdoor natural resources, possibly resulting in some similarities in comparison with animals from AF farms, whose diet consists exclusively of grazing acorns and grass in the *dehesa* [2,5]. The great dispersion of AF category in Figure 2 might be due to the high variability in resources grazed as well as the expression of dietary preferences by pigs [2], factors which could have led to a notable heterogeneity in the profile of the AF samples. 

Table 1 presents the different statistics obtained for calibration, cross-validation, and prediction in MPLS regressions. For the evaluation and comparison of the different regressions performed, it is important not only to consider the statistics concerning prediction or external validation, but also those related to calibration and cross-validation, as these are related to the robustness of the regressions. All the equations reached R^2^_C_ and R^2^_CV_ values higher than 0.92 and 0.85, respectively. The SEC and SECV values ranged between 0.05 and 0.19 (Table 1). Good statistics were also obtained in external validation (prediction), although slight differences were noticed depending on the evaluated spectral region: R^2^_p_ values between 0.79 and 0.94 and a classification success above 93.94% (Table 1). This success is consistent considering the absorbance differences identified after mean spectra comparison in Figure 1. The best performance was detected for the complete spectrum (400–2500 nm), with R^2^_P_ and SEP values of 0.89–0.94 and 0.13–0.17, respectively. The use of the other two regions implied a lower predictive performance, with an SEP of up to 0.25 and a medium R^2^_P_ of 0.84. However, although lower, these data are also considered successful results. The better results obtained by regressions using the spectral region of 400–2500 nm in comparison with those regressions performed using 850–2500 nm region may be related to the colour of the faeces, which seemed to be macroscopically different between both groups when processing. Probably due to the inclusion of compound feeds in the diet, FF samples tended to have a lighter brown colour than AF samples, which had a dark brown colour. The mathematical pre-treatment implied an improvement of models’ predictive performance in the three regions evaluated, specifically the use of derivatives; on the other hand, scatter correction did not obtain a significant improvement of MPLS regressions, probably associated with a stable base line of spectral data. As the descriptive statistics show (Table 1), the optimal spectral segment for the Iberian pig feeding regime prediction would be the complete visible and near infrared information (400–2500 nm), using a 1,5,5,1 mathematical treatment and no scatter correction, with an R^2^_P_ of 0.84, SEP of 0.13, and a classification success of 100%. 

In accordance with the findings highlighted by the PCA analysis (Figure 2), misclassified samples by regressions were concentrated in AF category. These samples were investigated and corresponded to those above-mentioned that, in Figure 2, overlapped with samples from OFF category. This fact is probably associated to the referred heterogeneity of AF feeding regime and the great deal of natural resources that pigs graze in the *dehesa* [2]. In this regard, although grass and acorns are predominant, previous studies have found the inclusion of up to 15 natural resources, depending on the individual, in the diet of AF Iberian pigs in the *dehesa* [2]. This could explain the heterogeneous spectral information obtained for the AF category, further complicating the model’s performance in comparison with FF feeding regime, where a higher homogeneity of diets is obtained because of the exclusive use of feed. The classification error and the overlapping between the AF and OFF categories have also been noticed in a similar study which employed faecal samples and chromatographic analysis to discriminate the Iberian pig diet [6]. Misclassified samples could also be related to the genetic variability of the animals included in the study, which has been suggested before as an influential factor, together with feeding system and individual variability, for the authentication of Iberian pig commercial categories using NIRS technology [15]. The present findings and the misclassification of samples from the AF group inside the FF group could also be related to the possible inclusion of feed in the feeding regime of the AF pigs [5]. Nowadays, there is some controversy over this inadequate strategy because, although it facilitates pigs’ handling, it is not established in regulation and final products are commercialised as acorn products without mentioning this information. What is more, it has not been deeply studied in the literature [6,21], so further studies would be welcomed in the future.

The methodology described using dummy variables in regressions has already been used for categorisation and authentication purposes in the Iberian pig sector, also obtaining successful classification results (73.8–93.4%) with NIRS analysis of four anatomical locations [15]. Despite the wide use of NIRS technology for evaluating Iberian pig products, the samples predominantly evaluated in the literature were usually collected postmortem, such as fresh meat or subcutaneous fat tissue [15,30,31,32]. In contrast, faecal samples allow an in vivo collection as no animal slaughtering would be required and does not imply animal immobilisation, because these can be collected non-invasively after spontaneous defaecation [16]. This last point is relevant in Iberian pig production, especially for extensive farms where sometimes animal handling is complicated. Therefore, analysis of faeces might stand as an innovative approach due to the advantages related to their collection and the information provided, which is directly related to the animal feeding and production system. In fact, the usefulness of NIRS analysis of pig faeces has already been demonstrated in other studies focused on nutrient digestibility evaluation [17,18,19]. 

The present approach may be appreciated in the sector as NIRS technology could complement the technicians’ field work, providing objective information about the pig feeding regime and therefore the resulting commercial categories, without any analytical information needed. Although the support given by NIRS technology would be of particular interest to those farms with outdoors facilities where fraudulent practices should be specially prevented, the present study includes a general comparison of feeding regimes in the sector (AF vs. FF) [5] in order to preliminarily evaluate NIRS feasibility for diet discrimination. Furthermore, if inspection visits are previously notified, animals fed with feed (OFF or IFF) can be fraudulently moved to field plots where AF are usually reared, so the discrimination proposed is considered meaningful. Another major advantage to emphasise about the approach presented is the great prediction accuracy achieved in this study using a mixed group of animals in terms of sex, live weight, and breed. Therefore, no constant conditions (such as an exact animal live weight) or specific models (depending on the sex or breed) would be necessary. This is considered essential for a potential application in the Iberian pig sector, which is characterised by a large variability regarding production systems and animal conditions. The potential inclusion of this type of methodology among certification procedures in the sector may be criticised due to possible consequences such as logistic drawbacks or price rises in Iberian products. It should however be noted that the availability of portable NIRS devices, which are relatively cheap and can be used directly on farms, would not significantly increase prices. The purchase of this type of equipment could be assumed for any accredited company. Apart from that, considering the uniqueness of the products coming from the Iberian pig, a slight increase in prices would probably be accepted by consumers if it results in good and strong guarantees, especially for those products coming from extensive systems.

The comparison used in the present study between AF and FF Iberian pigs has often been evaluated in the literature using other analytical techniques for preliminary evaluations and authentication purposes of the feeding regime of this breed [6,33,34,35]. In fact, this simplification of the current categorisation to only two groups has even been suggested by some authors as an option to clarify the market [36], since the current labelling may be confusing for customers [37,38]. In this regard, the results here presented would contribute to cover society’s increasing demand of valuable information about food authenticity and production systems [39], as the methodology used could support certification work in Iberian pig production and therefore in final products. 

Finally, although this preliminary study has highlighted interesting findings, additional research with higher sample throughputs from the different commercial categories and production systems would be welcome to develop definitive models. Breed purity discrimination stands as an interesting objective to be addressed in the near future due to its importance in the current framework of Iberian pig production, especially in the AF category [5]. In addition, the evaluation of how the faeces’ profile evolves over time when diet changes are performed in Iberian pig production should also be considered in forthcoming research and studies. 

## 4. Conclusions

The optimisation of analytical and objective strategies, which contribute to the authentication and certification of the Iberian pig’s feeding regime, is considered of paramount importance. In the present study, NIRS analysis of faeces has demonstrated a huge potential to perform an in vivo discrimination of the diet given to pigs on different farms. Different regression models using several spectral regions (400–2500, 850–2500, and 400–850 nm) and data pre-treatments (scatter correction and derivatives) were performed. A high level of accuracy was achieved (R^2^_P_ of 0.94 and classification success of 100%), thus confirming that this methodology can provide reliable and objective information to help in current certification procedures. Further work is in progress to increase the variability of the calibration and validation sets through the inclusion of samples from different farm handlings, animal characteristics, and diets, and to improve model robustness before implementation as a routine analysis. Specific regression models regarding the breed purity of Iberian pigs should also be evaluated in future studies because of its relevance in the market, as well as to its possible influence on the diet discrimination pursued in the present study. 

## Figures and Tables

**Figure 1 animals-14-01548-f001:**
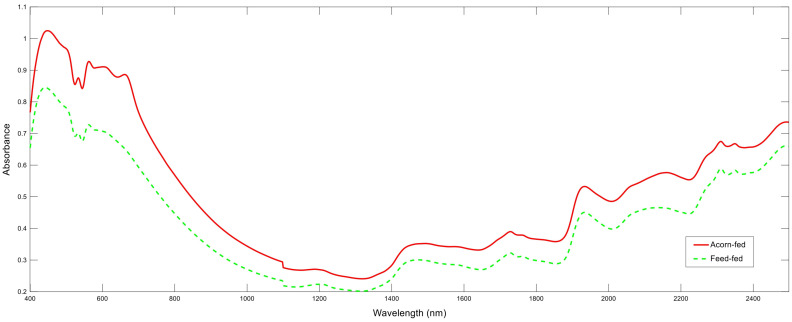
Representation of mean near infrared spectroscopy spectra of the two feeding groups evaluated.

**Figure 2 animals-14-01548-f002:**
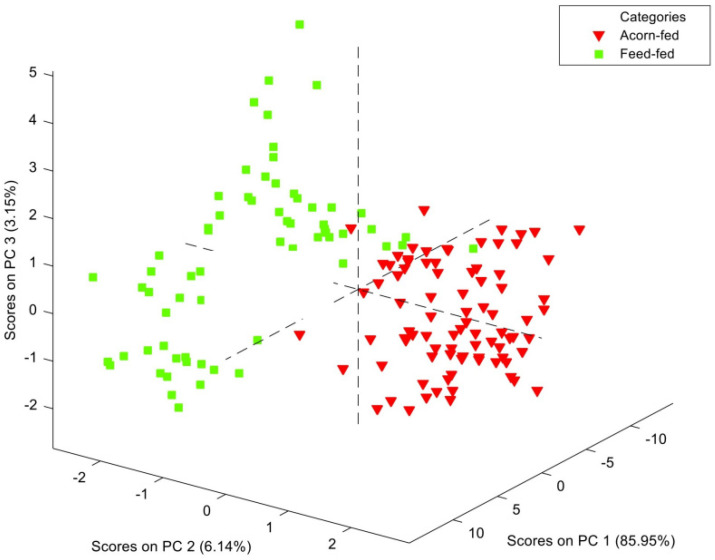
Score plot of principal component analysis performed with spectral data after near infrared spectroscopy analysis of faeces.

**Table 1 animals-14-01548-t001:** MPLS regression statistics of the equations performed for the classification of the Iberian pig feeding regime using spectral information from faeces.

Spectral Region (nm)	Data Pre-Treatment	Calibration and Cross-Validation						External Validation			
		N	PLS Terms	SEC	R^2^_C_	SECV	R^2^_CV_	N	SEP	R^2^_P_	Classification Success
400–2500	0, 0, 1, 1 none	129	12	0.11	0.95	0.14	0.92	33	0.16	0.91	100%
	1, 5, 5, 1 none	126	12	0.07	0.98	0.10	0.96	33	0.13	0.94	100%
	1, 5, 5, 1 SNV + DT	126	8	0.07	0.98	0.19	0.85	33	0.16	0.90	96.97%
	2, 5, 5, 1 none	130	10	0.07	0.98	0.13	0.93	33	0.15	0.92	100%
	2, 5, 5, 1 SNV + DT	127	5	0.10	0.96	0.19	0.85	33	0.17	0.89	100%
850–2500	0, 0, 1, 1 none	126	9	0.14	0.92	0.18	0.87	33	0.25	0.79	93.94%
	1, 5, 5, 1 none	129	11	0.08	0.97	0.11	0.95	33	0.18	0.89	100%
	1, 5, 5, 1 SNV + DT	121	11	0.06	0.99	0.08	0.97	33	0.19	0.87	96.97%
	2, 5, 5, 1 none	130	11	0.07	0.98	0.12	0.94	33	0.15	0.91	100%
	2, 5, 1, 1 SNV + DT	126	11	0.05	0.99	0.09	0.97	33	0.24	0.79	96.97%
400–850	0, 0, 1, 1 none	128	9	0.13	0.94	0.14	0.92	33	0.22	0.82	96.97%
	1, 5, 5, 1 none	127	9	0.09	0.97	0.12	0.94	33	0.24	0.78	96.97%
	1, 5, 5, 1 SNV + DT	128	11	0.10	0.96	0.16	0.90	33	0.21	0.83	96.97%
	2, 5, 5, 1 none	127	7	0.11	0.95	0.15	0.91	33	0.20	0.84	96.97%
	2, 5, 5, 1 SNV + DT	127	7	0.10	0.96	0.15	0.92	33	0.20	0.85	96.97%

N: number of samples used; SEC: standard error of calibration; R^2^_C_: coefficient of determination of calibration; SECV: standard error of cross-validation; R^2^_CV_: coefficient of determination of cross-validation; SEP: standard error of prediction; R^2^_P_: coefficient of determination of prediction; none: no scatter effect correction performed; SNV + DT: standard normal variate and de-trending transformation.

## Data Availability

Data are available upon request to Pablo Rodríguez-Hernández (v22rohep@uco.es) and with permission from study participants.

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
