# Peer review of "Regression Models for In Vivo Discrimination of the Iberian Pig Feeding Regime after Near Infrared Spectroscopy Analysis of Faeces"

_animals, 2024, doi:10.3390/ani14111548_

Round 1

Reviewer 1 Report

Comments and Suggestions for Authors

The study aims to explore Regression models for in vivo discrimination of the Iberian pig feeding regime after near infrared spectroscopy analysis of faeces.

Notes.

1. There are no detailed pig feeding rations. Add information on the composition and nutritional value of diets.

2. The title of the manuscript indicates regression models. But there are no regression models in the manuscript! Information needs to be agreed upon.

3. How was the research on the device calibrated? So I understand that this happened according to “wet” chemistry data? Where is the information about this in the manuscript?

4. The conclusion should be detailed. Too many common words!

Comments on the Quality of English Language

The manuscript is written in English.

There are some errors in the text.

The manuscript requires minor English editing.

Author Response

Reviewer 1

Firstly, we would like to thank the suggestions and comments from reviewers. We fully agree and think that these will help to improve the quality of the manuscript. The changes made in the manuscript have been introduced using Track Changes device from Microsoft Word in the version received from the editor.

  1. Comments and suggestions for authors
    • There are no detailed pig feeding rations. Add information on the composition and nutritional value of diets.

The suggestion of reviewer 1 has been considered and information related to the composition of nutritional value of diets has been added:

Changes:

  • Line 137-148: adding “Regarding the composition and nutritional value of diets compared in the present study, on one hand, the compound feeds used on OFF and IFF farms were based on different cereals, animal and vegetal fats, oil seeds and minerals as the main components. The mean average composition of feed used on farms was as follows: 10.82–13.72% crude protein, 3.23–7.64% crude fibre, 3.6–7.54% crude fat, 4.10–7.99% crude ash, 0.56–0.73% lysine, 0.16–0.24% methionine, 0.44–1.67% calcium, 0.35–0.7% phosphorus, and 0.13–0.26% sodium. On the other hand, and as mentioned above, the diet on AF farms was based on acorn and grass, although other natural resources could also be grazed by pigs in the dehesa, such as berries, roots, bushes or mushrooms [2]. The natural and diverse character of this diet substantially hampers the determination of a mean average composition, but an estimation about chemical composition of acorn and grass grazed by Iberian pigs during the montanera was published a few years ago [2,20].”
    • The title of the manuscript indicates regression models. But there are no regression models in the manuscript! Information needs to be agreed upon.

We would like to thank reviewer 1 for the comment. We may have not been very clear in this regard: PLS regression models have been developed using WINISI IV software to predict the value of the dummy variable "ALIM" to classify the samples into groups. However, the software does not provide the specific information of the resulting equation (coming from the regression model), just the statistics of the models (Table 1). The regression models are saved in a specific file with format .EQA and we cannot include that information in the manuscript, but rather only the statistics shown in the Table.

In general, papers dealing with the development of NIR chemometric PLS models do not show the model itself, but the statistics describing them. In fact, there are several studies published that also employ NIRS technology, this methodology (MPLS) and the word “model”, although the equations per se are not shown. An example is the following one: Horcada, A., Valera, M., Juárez, M., & Fernández-Cabanás, V. M. (2020). Authentication of Iberian pork official quality categories using a portable near infrared spectroscopy (NIRS) instrument. Food chemistry, 318, 126471.

However, if despite this explanation, reviewer 1 still thinks that title need to be changed, we would be willing to do it.

  • How was the research on the device calibrated? So I understand that this happened according to “wet” chemistry data? Where is the information about this in the manuscript?

We appreciate the comment by reviewer 1, and some clarification is needed at this point: NIRS technology is usually employed with quantitative approaches to predict the chemical composition of a product. For that purpose, reference data usually obtained through wet chemistry is used together with absorbance data to develop those predictive models. NIRS technology can also be used for qualitative purposes by developing discriminant models (PLS-DA, SIMCA, …), where only spectral absorbance data are used and there is no need to perform any chemical analyses.

The present paper aims at assigning the faecal samples into two groups (qualitative approach), but using a quantitative approach, the so-called Dummy variable technique. Hence, Modified Partial Least-Squares (MPLS) regressions have been developed where absorbance data are correlated with a dummy variable that represents a category (AF or FF), whose value is 1 for the samples belonging to category AF and 2 for the FF category samples. The MPLS model predicts the dummy variable value, and the samples belong to the category whose value is within ± 0.5.

This is the reason why we included in the “Results and discussion” section the following sentence: “The present approach may be appreciated in the sector as NIRS technology could complement the technicians’ field work, providing objective information about the pig feeding regime and therefore the resulting commercial categories, without any analytical information needed”. MPLS technique used this way allows us to discriminate samples by predicting a dummy variable but without any analytical or lab data needed.

  • The conclusion should be detailed. Too many common words!

We agree with reviewer 1 and some new information has been added to the “Conclusions” section in order to achieve a clearer and detailed explanation.
Changes:

  • Lines 375-376: adding “Different regression models using several spectral regions (400-2500, 850-2500 and 400-850 nm) and data pre-treatments (scatter correction and derivatives) were performed.”
  • Line 377: removing “by the chemometric models performed”.
  • Line 378: adding “[…] and classification success of 100% […]”

  1. Comments on the Quality of English Language
    • The manuscript is written in English. There are some errors in the text. The manuscript requires minor English editing.

The whole manuscript has been revised for English improvement. A native speaker has suggested several corrections and minor changes related to written English have been introduced (marked with a yellow background).

Reviewer 2 Report

Comments and Suggestions for Authors

The manuscript entitled "Regression models for in vivo discrimination of the Iberian pig feeding regime after near infrared spectroscopy analysis of faeces" presents an interesting objective.

The focus of the study is to define analytical methodologies aimed at certifying the pig feeding .

The procedure is rather complex, however Authors were able to use some statistical approaches and database analyzes in order to identify the constituent components of feces.

However, I advise the authors to be clearer in presenting the results, with a more informative approach.

Author Response

Reviewer 2

Firstly, we would like to thank the suggestions and comments from reviewers. We fully agree and think that these will help to improve the quality of the manuscript. The changes made in the manuscript have been introduced using Track Changes device from Microsoft Word in the version received from the editor.

  1. Comments and suggestions for authors
    • However, I advise the authors to be clearer in presenting the results, with a more informative approach.

We would like to thank reviewer 2 for this comment. We agree and the presentation of results has been revised in order to obtain a comprehensive and precise explanation. Several changes have been introduced in the manuscript in this regard.

Changes:

  • Line 223: adding a new citation of Figure 1 in order to obtain a clearer vision of what is being explained in that part of the manuscript.
  • Line 236: adding a brief explanation for outlier concept; “[…] or atypical samples […]”.
  • Lines 242-243: two clarifications related to scores’ colour in PCA graphic representation have been introduced in order to obtain an easier interpretation of Figure 2.
  • Lines 256-259: adding “For the evaluation and comparison of the different regressions performed, it is important not only to consider the statistics concerning prediction or external validation, but also those related to calibration and cross-validation, as these are related to the robustness of the regressions.”.
  • Lines 268-269: adding “However, although lower, these data were also considered successful results.”.
  • Lines 277-278: adding “[…] probably associated with a stable base line of spectral data.”.
  • Lines 290-293: the sentence has been rewritten in order to obtain a clearer explanation.

Reviewer 3 Report

Comments and Suggestions for Authors

- The manuscript is well written and the results and discussion are straight forward

- Citation and references are not according to journal Style.

- It seems nice to have a sophisticated analysis for certification, however the authors missed out that the cost for certification or iberian pig products may increase, official sample taking, analysis cost, accreditation…etc This should be probablymentioned somewhere in the manuscript

If animals are only fed with acorn for the last month can this be differentiated by this faeces method if animals were fed before without acorn?? Or does this analysis needs to be done several time to get a clear picture of life time feeding. Otherwise it only gives information of the last feeding situation

Author Response

Reviewer 3

Firstly, we would like to thank the suggestions and comments from reviewers. We fully agree and think that these will help to improve the quality of the manuscript. The changes made in the manuscript have been introduced using Track Changes device from Microsoft Word in the version received from the editor.

  1. Comments and suggestions for authors
    • Citation and references are not according to journal Style.

The comment of reviewer 3 has been considered and the citation style has been changed throughout the whole manuscript, using the specific style of Animals.

  • It seems nice to have a sophisticated analysis for certification, however the authors missed out that the cost for certification or iberian pig products may increase, official sample taking, analysis cost, accreditation…etc This should be probably mentioned somewhere in the manuscript.

We would like to thank reviewer 3 for this interesting comment. We think that the inclusion of this information is important and will help to improve the quality of the manuscript, as these are possible consequences resulting from the application of this type of methodology in the sector. However, it is important to highlight that the inspection visits are mandatory nowadays, so no additional costs will be incurred. One of the advantages of NIR technology is that once the instrument is calibrated, the cost of analysing samples is close to zero. Also, the possibility of employing portable NIRs equipment, which is relatively cheap and can be transported to farms, would not significantly increase prices. Apart from that, considering the uniqueness of the products coming from the Iberian pig, a slight increase in price would be accepted by consumers if it results in good, strong guarantees.

New sentences have been added to introduce this information:

  • Lines 341-351: “The potential inclusion of this type of methodology among certification procedures in the sector may be criticised due to possible consequences such as logistic drawbacks or price rises in Iberian products. It should however be noted that technical visits are already mandatory nowadays for visual and documentary inspections, so this will not result in any additional expenditure. Furthermore, the availability of portable NIRS devices, which are relatively cheap and can be used directly on farms, would not significantly increase prices. The purchase of this type of equipment could be assumed for any accredited company. Apart from that, considering the uniqueness of the products coming from the Iberian pig, a slight increase in price would probably be accepted by consumers if it results in good and strong guarantees, especially for those products coming from extensive system”.
    • If animals are only fed with acorn for the last month can this be differentiated by this faeces method if animals were fed before without acorn?? Or does this analysis needs to be done several time to get a clear picture of life time feeding. Otherwise it only gives information of the last feeding situation.

We appreciate this reflection by reviewer 3. We think it is a relevant question.  When we started to plan this type of study using faeces, we took this into account. In this regard, the diet given to pigs results in notable changes in faeces’ profile relatively quickly. In other words, a change of diet would be noted in faeces only after a few days. Therefore, this condition fits in the approach we were looking for. This type of analysis, NIRS analysis of faeces, gives us information about the last feeding situation, which is what the technician during the inspection visits (which are mandatory by regulation) try to certificate or discriminate. Moreover, farmers are not informed in advance to avoid fraudulent diet changes.

However, further research would be interesting in order to evaluate how faeces’ profile change over time in this specific case: the Iberian pig. For future studies the same groups of pigs could be continuously evaluated from some days before starting the montanera, when the diet is still based on compound feed, to some days after this change of diet, when the animals start to eat acorn, grass and natural resources. Thus, we would be able to accurately know for how many days NIRS technology gives information.

Considering the above-mentioned background, a new sentence has been added in the final part of the manuscript:

Lines 367-369: “In addition, the evaluation of how faeces’ profile evolves over time when changes of diets are performed in Iberian pig production should also be considered in forthcoming evaluations and studies.”.

Reviewer 4 Report

Comments and Suggestions for Authors

Interesting topic as it could be a useful method for certification in this market.

Line 11- consider “indigenous” or “native” or other more common word than “autochthonous”

Line 12: Different productive systems and feeding regimes are considered by regulation- This is unclear- Different rearing practices and feeding regimes are regulated resulting in different labeling schemes? Consider rephrasing.

Line 22- "were employed considering their easy collection,"- odd wording. Were used due to ease of collection?

Line 36 and throughout - dehesa  should be capitalized

Line 40- despite of the important relation between- despite “the”,

Line 41- the current productive systems in Iberian pig sector- the production systems in the Iberian pig sector (also should replace productive systems with “production systems” and Iberian pig sector with “the Iberian pig sector” throughout the text).

Line 44- What is meant by “ecosystem services”? - consider clarifying. not sure the reader will understand what this means.

Line 58- Contemplated by- This seems to be the wrong word. Accepted by? Employed by?

Line 68- “considering the price and reputation of products coming from Iberian pig production.” The inspection methodology is potentially flawed because it is subjective and could be manipulated. The accuracy of the certification is important because consumers pay a high price for these products so they should be assured that they are receiving what they pay for. This sentence could be more clear. Consider re-phrasing.

Line 70- “Against that background, a great deal of research focused on the development and evaluation of this type of equipment and techniques has been published in recent years in the sector” May want to add what type of techniques have been looked at to clarify that NIRS is something new that is investigated here.

Line 99- “with approximately from” – from approximately

Line 101- “months age” – months of age

Line 101- “Racial purity of animals varied” – The word “racial” has negative connotations. Consider using the word “breed” or “genetic” here and line 333

Line 102- “to the crossbred” – Due to crossbreeding with the Duroc breed.

Line 106- “A first group of extensive farms belonging to AF commercial category, including grazing animals exclusively fed with natural resources: mainly acorns and grass in the dehesa, without using any supplementary feed (Rodríguez-Estévez et al., 2009).” This is not a complete sentence. Change belonging to belonged or , including to included

Line 111- “On the other hand, in the present study” consider removing, not needed

Line 111- “farms identified” – were identified

Line 135- “introduced in refrigeration boxes and” – were refrigerated and

Should split the results from the discussion. This is hard for the reader to understand and not required by the journal.

Line 266- “This fact is probably associated to the above-mentioned heterogeneity of AF feeding regime” it would help the reader if it were briefly noted again why AF (particularly OFF) might be more heterogeneous since it is hard to keep track of the distinctions between the categories.

Line 290- “successful classification results (73.8-93.4%)” classification of what?

Line 301- “the usefulness of NIRS analysis of pig faeces has already been demonstrated” this makes it sound like this study has already been done so it would be good to add what NIRS has been demonstrated to be useful for.

Line 307- “Although the support given by NIRS technology would be of particular interest in those farms with outdoors facilities, the present study includes a general comparison of feeding regimes in the sector (AF vs. FF) (Ministerio de Agricultura Alimentación y Medio Ambiente, 2014) in order to preliminarily evaluate NIRS feasibility for diet discrimination. Furthermore, animals fed with feed (OFF or IFF) can be fraudulently moved prior to certification visits to field plots where AF are usually reared, so the discrimination proposed is considered meaningful.” This is confusing but an important point. Why is it of interest to outdoor facilities? How is diet discrimination related to people fraudulently moving animals prior to the certification visit?

Consider including more on how this method would specifically prevent fraudulent activities during the certification process. What specific practices would the use of NIRS prevent? Could people just switch the diet a few days prior to the visit and NIRS would certify them? Or is that an area that needs for more study?

Line 320- “The comparison between AF and FF Iberian pigs used” more information on what was looked at in those studies and how it differs from this study would help the reader.

Line 323- “In fact, this simplification of the current categorization to only two groups has been even suggested by some authors as an option to clarify the market” Please discuss how NIRS can only differentiate diet and not the other components of the labeling scheme such as free range and other variables (OFF vs IFF) because this seems like a disservice to people who think they are paying for a quality product that also comes with enhancements to other production practices, not just a way to “clarify the market”

Comments on the Quality of English Language

The English is not bad but is somewhat confusing in place. I tried to address those areas that I found in my review. 

Author Response

Reviewer 4

Firstly, we would like to thank the suggestions and comments from reviewers. We fully agree and think that these will help to improve the quality of the manuscript. The changes made in the manuscript have been introduced using Track Changes device from Microsoft Word in the version received from the editor.

  1. Comments and suggestions for authors
    • Line 11- consider “indigenous” or “native” or other more common word than “autochthonous”.

The comment made by reviewer 4 has been considered and “autochthonous” has been changed to “native” throughout the whole manuscript.

  • Line 12: Different productive systems and feeding regimes are considered by regulation- This is unclear- Different rearing practices and feeding regimes are regulated resulting in different labeling schemes? Consider rephrasing.

 We would like to thank the suggestion by reviewer 4, we think it is clearer now. The sentence has been rephrased.

  • Line 22- "were employed considering their easy collection,"- odd wording. Were used due to ease of collection?

Faeces samples were selected for several reasons. Apart from the great deal of information this type of sample provides about the individual metabolism and homeostasis, also for logistic advantages, especially the ease of collection in farms. This strength is highlighted on extensive farms, where animals freely graze in large extensions.

  • Line 36 and throughout - dehesa should be capitalized

We appreciate the suggestion made by reviewer 4 but, after revising many articles published in the literature, the word “dehesa” is not usually capitalized. Therefore, it has not been changed.

  • Line 40- despite of the important relation between- despite “the”,

We would like to thank reviewer 4 this comment, the word “of” has been deleted.

  • Line 41- the current productive systems in Iberian pig sector- the production systems in the Iberian pig sector (also should replace productive systems with “production systems” and Iberian pig sector with “the Iberian pig sector” throughout the text).

The suggestions of reviewer 4 have been considered and both expressions have been changed throughout the whole manuscript: “productive systems” by “production systems”, and “Iberian pig sector” by “the Iberian pig sector”.

  • Line 44- What is meant by “ecosystem services”? - consider clarifying. not sure the reader will understand what this means.

We would like to thank the comment, perhaps it is not at all clear. To avoid any confusion for the reader, the sentence mentioned by reviewer 4 has been deleted.

  • Line 58- Contemplated by- This seems to be the wrong word. Accepted by? Employed by?

We have considered the comment of reviewer 4 and “contemplated by” has been replaced by “established in/by”.

  • Line 70- “Against that background, a great deal of research focused on the development and evaluation of this type of equipment and techniques has been published in recent years in the sector” May want to add what type of techniques have been looked at to clarify that NIRS is something new that is investigated here.

We think that the inclusion of the information suggested by reviewer 4 is important. Therefore, new information has been added:

  • Lines 85-86: adding “[…] such as methodologies based on liquid and gas chromatography, or different spectroscopic techniques.”.
    • Line 99- “with approximately from” – from approximately

We agree with the comment of reviewer 4. The sentence has been changed.  

  • Line 101- “months age” – months of age

We appreciate this comment. The word “of” has been removed.

  • Line 101- “Racial purity of animals varied” – The word “racial” has negative connotations. Consider using the word “breed” or “genetic” here and line 333

We have considered the suggestion of reviewer 4 and the word “breed” has been used instead of “racial”. This change has been introduced in Lines 113, 361 and 380.

  • Line 102- “to the crossbred” – Due to crossbreeding with the Duroc breed.

We agree with this change and we have included it. The sentence is now clearer.

  • Line 106- “A first group of extensive farms belonging to AF commercial category, including grazing animals exclusively fed with natural resources: mainly acorns and grass in the dehesa, without using any supplementary feed (Rodríguez-Estévez et al., 2009).” This is not a complete sentence. Change belonging to belonged or , including to included

We appreciated this comment, the sentence may be quite confusing. The suggestions of reviewer 4 have both been included.

  • Line 111- “On the other hand, in the present study” consider removing, not needed

Part of the sentence indicated by reviewer 4 has been removed (“[…] in the present study […]”).

  • Line 111- “farms identified” – were identified

We agree with this suggestion. The word “were” has been included in the sentence.

  • Line 135- “introduced in refrigeration boxes and” – were refrigerated and

We also agree with this suggestion and the sentence has been rewritten.

  • Should split the results from the discussion. This is hard for the reader to understand and not required by the journal.

We would like to thank reviewer 4 for this suggestion. We often separate results and discussion in some of our papers, but in this case we truly believe that the comprehension of the content is better with the current form. We think that the discussion of results directly, once these are presented, contributes to a better understanding of what we want to present.

We hope reviewer 4 understands our position.

  • Line 266- “This fact is probably associated to the above-mentioned heterogeneity of AF feeding regime” it would help the reader if it were briefly noted again why AF (particularly OFF) might be more heterogeneous since it is hard to keep track of the distinctions between the categories.

We agree with reviewer 4, so we have included additional information in that sentence to help the reader:

  • Lines 292-293: adding “[…] and the great deal of natural resources that pigs graze in the dehesa […]”.
    • Line 290- “successful classification results (73.8-93.4%)” classification of what?

The data included in that sentence refers to the correct classification of pig’ carcasses depending on the commercial category (the aim of the study).

  • Line 301- “the usefulness of NIRS analysis of pig faeces has already been demonstrated” this makes it sound like this study has already been done so it would be good to add what NIRS has been demonstrated to be useful for.

We totally agree with reviewer 4 and the sentence has therefore been extended to include information about what NIRS has been demonstrated to be useful for:

  • Line 324-325: adding “ […] focused on nutrient digestibility evaluation […]”
    • Line 307- “Although the support given by NIRS technology would be of particular interest in those farms with outdoors facilities, the present study includes a general comparison of feeding regimes in the sector (AF vs. FF) (Ministerio de Agricultura Alimentación y Medio Ambiente, 2014) in order to preliminarily evaluate NIRS feasibility for diet discrimination. Furthermore, animals fed with feed (OFF or IFF) can be fraudulently moved prior to certification visits to field plots where AF are usually reared, so the discrimination proposed is considered meaningful.” This is confusing but an important point. Why is it of interest to outdoor facilities? How is diet discrimination related to people fraudulently moving animals prior to the certification visit?

Firstly, the methodology is of particular interest for outdoor facilities because fraudulent practices are more common on this type of farm. An intensive farm with close buildings and without outdoor facilities cannot pretend to be certified as OFF or AF. Also, we wrote that that methodology would be of particular interest in outdoor facilities because this type of farm would be where a portable NIRS device would be more useful, as the animals have freedom of movement. Conversely, indoor facilities allow easier access to animals.

Regarding the fraudulent move of animals prior to certification visits, this happens when inspection visits are previously notified to the farmers, so some animals can be moved to those plots where AF pigs graze.

To achieve a clearer explanation, some changes have been introduced in this part of the manuscript:

  • Lines 330-331: adding “[…] where fraudulent practices should be specially prevented […].
  • Lines 335: adding “[…] if inspection visits are previously notified […]”.
    • Consider including more on how this method would specifically prevent fraudulent activities during the certification process. What specific practices would the use of NIRS prevent? Could people just switch the diet a few days prior to the visit and NIRS would certify them? Or is that an area that needs for more study?

The methodology here described would be useful to detect sudden changes of diet in Iberian pigs prior to certification visits which try to obtain a more expensive labelling, as NIRS analysis of faeces gives information about the last feeding situation. Changes of diet would be noted in faeces’ profiles only after a few days.

However, and as we have answered reviewer 3, further research would be interesting in order to evaluate how faeces’ profiles change over time. For future studies, the same groups of pigs could be continuously evaluated from some days before starting the montanera, when the diet is still based on compound feed, to some days after this change of diet, when the animals start to eat acorn, grass and natural resources. Thus, we would be able to accurately know for how many days NIRS technology gives information.

  • Line 320- “The comparison between AF and FF Iberian pigs used” more information on what was looked at in those studies and how it differs from this study would help the reader.

We would like to thank this comment, we agree that it should be clarified. Therefore, the sentence has been rewritten and extended to achieve a clearer form.

  • Line 323- “In fact, this simplification of the current categorization to only two groups has been even suggested by some authors as an option to clarify the market” Please discuss how NIRS can only differentiate diet and not the other components of the labeling scheme such as free range and other variables (OFF vs IFF) because this seems like a disservice to people who think they are paying for a quality product that also comes with enhancements to other production practices, not just a way to “clarify the market”

The information provided by the methodology here described is not related exclusively to the diet and the feeding regime because a faeces’ profile is affected by several variables, not only the diet (although it would be the main one). The facilities and therefore the exercise/movement of animals, genetic constitution (which is highlighted in the final part of the manuscript to be addressed in the future), or even the sex are examples of variables which can influence the faeces’ profile. This type of sample provides a great deal of information.

The clarification of the market which is mentioned in that part of the manuscript is exclusively related to the decrease of commercial categories in terms of animal diet, from three (AF, OFF, IFF) to only two (AF and FF).

  1. Comments on the Quality of English Language
    • The English is not bad but is somewhat confusing in place. I tried to address those areas that I found in my review.

We would like to thank reviewer 4 for all the recommendations, including those related to the written English. We think these have improved the manuscript quality. Also, a native speaker has reviewed the whole text and has proposed several changes, which have been included and are marked with a yellow background.

Round 2

Reviewer 3 Report

Comments and Suggestions for Authors

no further objections

Comments on the Quality of English Language

no further objections

Author Response

We would like to thank these last suggestions and comments from reviewer 3. We fully agree with them.

As in the previous version, changes made in the manuscript have been introduced using Track Changes device from Microsoft Word.

Reviewer 4 Report

Comments and Suggestions for Authors

Paper is much improved especially to give more background to help people understand the use of the technology for this purpose. Language is also more refined and reads well. A few minor errors are noted.

Line 112- On the other hand, "a" second

Line 148- has already been already used (removed one "already")

Line 334- Since the testing has a cost it seems wrong to say that there won't be any additional expenditures. May want to clarify that the auditing authorities would likely assume any additional testing costs but that fees for audits may increase for farmers if that is indeed the case.

Author Response

Once again, we would like to thank these last suggestions and comments from reviewer 4. We fully agree with them.

As in the previous version, changes made in the manuscript have been introduced using Track Changes device from Microsoft Word.

Reviewer 4

Paper is much improved especially to give more background to help people understand the use of the technology for this purpose. Language is also more refined and reads well. A few minor errors are noted.

  1. Comments and suggestions for authors
    • Line 112- On the other hand, "a" second.

We would like to thank reviewer 4 for this comment, we have added “a”.

  • Line 148- has already been already used (removed one "already")

We have considered this comment and one “already” has been removed.

  • Line 334- Since the testing has a cost it seems wrong to say that there won't be any additional expenditures. May want to clarify that the auditing authorities would likely assume any additional testing costs but that fees for audits may increase for farmers if that is indeed the case.

We would like to thank reviewer 4, we agree with this comment. That part of the paragraph has been modified and the information related to the absence of additional expenditure has been removed (as maybe a slight increase of prices would be possible).

The use of a portable NIRS equipment would not imply huge differences of prices, as these are relatively cheap and the amortisation cost (which would represent the increase of audits’ fees) would be low. We have let this information in the paragraph.